# Evaluating the Dissemination and Implementation of a Community Health Worker-Based Community Wide Campaign to Improve Fruit and Vegetable Intake and Physical Activity among Latinos along the U.S.-Mexico Border

**DOI:** 10.3390/ijerph19084514

**Published:** 2022-04-08

**Authors:** Paul Gerardo Yeh, Belinda M. Reininger, Lisa A. Mitchell-Bennett, Minjae Lee, Tianlin Xu, Amanda C. Davé, Soo Kyung Park, Alma G. Ochoa-Del Toro

**Affiliations:** 1Division of Health Promotion & Behavioral Sciences, Brownsville Regional Campus, School of Public Health, University of Texas Health Science Center, 80 Fort Brown, Brownsville, TX 78520, USA; belinda.m.reininger@uth.tmc.edu (B.M.R.); lisa.mitchell-bennett@uth.tmc.edu (L.A.M.-B.); 2Department of Physician Assistant, College of Health Professions, University of Texas Rio Grande Valley, 1201 West University Blvd., Edinburg, TX 78539, USA; 3Postdoctoral Fellow, National Cancer Institute Cancer Control Research Training Program, School of Public Health, University of Texas Health Science Center, 1200 Pressler Street, Houston, TX 77030, USA; 4Hispanic Health Research Center, School of Public Health, University of Texas Health Science Center, 1 West University Blvd., Brownsville, TX 78520, USA; amanda.c.dave@uth.tmc.edu (A.C.D.); alma.g.ochoadeltoro@uth.tmc.edu (A.G.O.-D.T.); 5Division of Biostatistics, Department of Population & Data Sciences, University of Texas Southwestern Medical Center, 5323 Harry Hines Blvd., Dallas, TX 75390, USA; minjae.lee@utsouthwestern.edu; 6Department of Biostatistics and Data Science, School of Public Health, University of Texas Health Science Center, 1200 Pressler Street, Houston, TX 77030, USA; tianlin.xu@uth.tmc.edu (T.X.); soo.kyung.park@uth.tmc.edu (S.K.P.)

**Keywords:** community-wide campaign, community health worker, physical activity, behavioral dietary intervention, built environment, implementation science, dissemination research, health behavior promotion, Latino community health, U.S.-Mexico border health

## Abstract

This study evaluated the dissemination and implementation of a culturally tailored community-wide campaign (CWC), *Tu Salud* ¡*Si Cuenta*! (TSSC), to augment fruit and vegetable (FV) consumption and physical activity (PA) engagement among low-income Latinos of Mexican descent living along the U.S.-Mexico Border in Texas. TSSC used longitudinal community health worker (CHW) home visits as a core vehicle to enact positive change across all socioecological levels to induce behavioral change. TSSC’s reach, effectiveness, adoption, implementation, and maintenance (RE-AIM) was examined. A dietary questionnaire and the Godin-Shepherd Exercise Questionnaire measured program effectiveness on mean daily FV consumption and weekly PA engagement, respectively. Participants were classified based on CHW home visits into “low exposure” (2–3 visits) and “high exposure” (4–5 visits) groups. The TSSC program reached low-income Latinos (*n* = 5686) across twelve locations. TSSC demonstrated effectiveness as, compared to the low exposure group, the high exposure group had a greater FV intake (mean difference = +0.65 FV servings daily, 95% CI: 0.53–0.77) and an increased PA (mean difference = +185.6 MET-minutes weekly, 95% CI: 105.9–265.4) from baseline to the last follow-up on a multivariable linear regression analysis. Multivariable logistic regression revealed that the high exposure group had higher odds of meeting both FV guidelines (adjusted odds ratio (AOR) = 2.03, 95% CI: 1.65–2.47) and PA guidelines (AOR = 1.36, 95% CI: 1.10–1.68) at the last follow-up. The program had a 92.3% adoption rate, with 58.3% of adopting communities meeting implementation fidelity, and 91.7% of communities maintaining TSSC. TSSC improved FV consumption and PA engagement behaviors among low-income Latinos region wide. CHW delivery and implementation funding positively influenced reach, effectiveness, adoption, and maintenance, while lack of qualified CHWs negatively impacted fidelity.

## 1. Introduction

American Latinos have lower rates of fruit and vegetable (FV) consumption [1,2] and physical activity (PA) [3,4,5] compared to other ethnic groups, and thus disproportionately do not meet the guidelines of consuming five servings of FV daily [6], nor achieving at least 150 min of moderate and vigorous PA weekly [7,8]. Such behavioral disparities are related to factors across all socioecological levels, ranging from the individual-level lack of self-efficacy [9] up to environmental factors such as the lack of PA infrastructure [3]. This behavioral discrepancy results in increased susceptibility [10,11,12,13,14] and mortality [15] to noncommunicable diseases (NCDs). Therefore, there is a dire public health need to disseminate evidence-based health behavior promotion interventions that address multilevel ecological factors for American Latinos.

### 1.1. Utilizing the Community-Wide Campaign Model for Health Behavior Augmentation

A cost-effective and evidence-based manner to address health behaviors are community-wide campaigns (CWC), which take a holistic approach to target factors influencing behavior across all socioecological levels, ranging from individualized risk factor screening to interpersonal social support to environmental infrastructure improvements [16,17]. CWCs can be tailored to a community by harnessing community assets to implement the CWC intervention [9,17,18]. Some CWC interventions use the Transtheoretical Model [19,20,21] or Social Cognitive Theory [22,23,24] to tailor strategies targeting factors at the individual-level [25] to the community and policy level [26,27]. However, CWCs targeting PA are novel and are generally not based on behavioral theory [28,29]. Moreover, CWCs globally are focused on high-income countries, with CWCs rarely being implemented in low-resource communities [26] that generally face more socioeconomic barriers and health disparities [15,25]. One notable exception is the *Tu Salud* ¡*Si Cuenta*! (Your Health Matters!) (TSSC) program, which adapts a CWC model for Latinos of Mexican descent living along the U.S.-Mexico border in Texas [1,18,30]. In its pilot phase, TSSC Latino participants in one intervention city demonstrated increased odds of meeting PA guidelines compared to Latinos in a control border city [17]. TSSC’s programming is described in detail below.

### 1.2. Dissemination and Implementation of Community-Wide Campaigns

While CWCs are effective in improving outcomes, the adoption and scale up of these programs at the population level can take decades [31]. Dissemination and implementation research enhances the spread of evidence-based interventions to priority populations to mitigate the gap between knowledge of what can maximize healthy outcomes and what is being delivered in community settings [32,33]. A research gap exists in the real-world implementation of CWC programs [25], with a particular lack of studies focused on Latino populations. To date, with the exception of the North Karelia (Finland) study, which was later nationally disseminated [34,35], there has not been a study specifically examining the public health impact of the broad dissemination of a CWC [36]. This study thus evaluates the real-world dissemination of the evidence-based TSSC program to 12 locations on the U.S.-Mexico border as a novel health behavior promotion effort for the region.

### 1.3. Research Questions

The translational science-derived RE-AIM (Reach, Effectiveness, Adoption, Implementation, and Maintenance) framework provides a structured manner to evaluate the public health impact of an intervention’s dissemination and informed future program implementation and adaptation through iterative reflection [37]. Accordingly, this study utilized the RE-AIM analytical framework [37] for its purpose of quantitatively evaluating the public health impact of the dissemination and implementation of TSSC’s regional expansion to 12 municipalities along the U.S.-Mexico border from January 2014–November 2019. Our research questions included: (1) Reach—what percent of the population was reached by TSSC’s community health worker (CHW) home visits and were our priority population of low-income Latinos? (2) Effectiveness—is the TSSC program associated with improvements in FV consumption and PA engagement levels among participants across and by location? (3) Adoption—what percent of communities offered the TSSC program adopted it? (4) Implementation—what percent of locations implemented the program with fidelity? and (5) Maintenance—what percent of participating locations maintained the program at the end of the study period?

We hypothesized that adapting a CWC model would appeal to and reach our priority population, that with increasing CHW home visits there would be significantly increased levels of FV consumption and PA engagement aggregately and by location, that most locations offered the program would adopt it given the TSSC’s prior successes in health behavior uptake [17,18], and that implementation and maintenance would be high given the incorporation and training of community CHWs in all participating locations.

## 2. Materials and Methods

### 2.1. Study Design and Setting

With the goal of combining an effectiveness study and implementation research, we conducted an Effectiveness-Implementation Hybrid Type I study design [38]. To evaluate TSSC’s effectiveness, we examined participants’ FV consumption and PA level in a one-group, pretest-posttest quasi-experimental design, where each participant’s baseline values of self-reported FV consumption and PA engagement were compared to self-reported FV consumption and PA at the last CHW home visit. As the TSSC intervention had shown evidence of efficacy in one local community [17,18], this quasi-experimental design was used to disseminate and implement TSSC regionally to reach as many Latino adults as possible who may benefit from the intervention [39]. There was no control group in this study, and local municipalities were not randomized to the TSSC intervention. All Latino participants eligible for CHW home visits (criteria detailed below) with a baseline and at least one follow-up measure of FV consumption and PA were included in this study to evaluate TSSC’s effectiveness. Moreover, given the potential variability in implementation by municipalities, we used the RE-AIM framework to more holistically assess the dissemination and implementation of the scaled-up TSSC program. Therefore, the dual focus of behavioral outcomes for effectiveness with a broader exploration of TSSC’s dissemination and implementation made this overall study an Effectiveness-Implementation Hybrid Type I study [38]. Such study designs can support more rapid translation of research findings into practical information to benefit communities [38].

TSSC took place in the U.S.-Mexico border region of the lower Rio Grande Valley, which is young (mean age of 45 years) [40] and rapidly growing [41], with a 2010 population of over 1.3 million, of which over 93% are Latino and 87% are of Mexican descent [41]. This area has over one-third of its population in poverty [42,43], with a per capita income of half the American average [41]. As a persistent poverty region, health disparities abound [43]. 28% of adults in the region have diabetes, and 84% of adults are obese or overweight [44]. 67% of the adult population lacks health insurance [40]. Given these poor socioeconomic indicators, the uptake of preventive health behaviors of FV consumption and PA engagement is disproportionately low locally [9,15], even relative to the national Latino population, who already have a low uptake of health behaviors [2,3,4,5]. The lowest rate of PA engagement are seen in those in poverty [8,45], with NCDs like obesity [46], and being of a female sex [46]. In 2011, prior to TSSC, there was a 33.3% compliance rate with PA guidelines for Latinos in the Rio Grande Valley compared to 44.1% for Latinos nationally (*p* = 0.019) and a 14.8% compliance rate with FV guidelines compared to 21.93% for Latinos nationally (*p* < 0.0001) [13]. We thus wanted to examine TSSC’s ability to enhance FV and PA behaviors, particularly in poor Latinos with NCDs easily screened for in a non-clinical CHW home visit setting (i.e., obesity and hypertension).

#### 2.1.1. Implementing Locations

From 2014–2019, the TSSC program was disseminated and implemented in 12 locations in Texas’ Rio Grande Valley, neighboring Mexico. The location included two county precincts (“Precinct”) and two urban areas (“City”) with a population of over 75,000 each, two small towns with a population of 10,000 to 70,000 (“Town”), and six rural areas each with populations <5000 (“Rural”) [47] as shown on Figure 1.

#### 2.1.2. Tu Salud ¡Si Cuenta! Program Description and Implementation over Time

TSSC culturally tailored the CWC model to the U.S.-Mexico border Latino population to address their lack of social support and community- and environmental-level facilitators for healthy behaviors [30]. TSSC harnesses the unique South Texas community asset of community health workers (CHWs), or *promotoras*, who are lay neighborhood-based health workers trained to provide health information to their community to combat systemic healthcare inaccessibility [48,49]. CHWs use their shared culture, community, and life experiences to connect with the focal population [9,18,30]. CHWs are gatekeepers to the acceptability and diffusion of health promotion efforts in Latino communities [50,51], with CHW home visits augmenting the uptake of health behaviors in Latinos [9,13,17,30] and instigating intergenerational changes in behavior within households [52]. TSSC used existing community CHWs, paid by the local municipality they live in, to deliver home visits to TSSC participants. CHWs also had monthly trainings, detailed elsewhere [30], occurring prior to and during the TSSC intervention, and included rapport-building education to enhance participant receptiveness to the CHW home visits, review of home visit educational materials to ensure programmatic consistency, and motivational interviewing training to help facilitate participant lifestyle behavior change efforts [52].

The TSSC recruitment and enrollment process is presented in Figure 2. Through community outreach, we recruited individuals who agreed to have an initial CHW visit for baseline risk factor screening (body mass index (BMI) and blood pressure (BP)) (Figure 2). Every recruited individual received an initial CHW home visit for risk factor screening. However, a goal of TSSC’s CHW home visits was to mitigate disease morbidity and mortality in participants with NCDs that can be screened for and monitored in a noninvasive manner by CHWs during home visits, which was obesity and hypertension [18]. We thus enrolled only participants with abnormal BMI and/or BP to the follow-up CHW home visits, and the vast majority of our participants were eligible (Figure 2). Those with follow-up CHW home visits formed the sample of TSSC participants of interest in this study (Figure 2). Any individual, regardless of vitals screening status, could engage in other TSSC programming components (Figure 2).

CHW home visit procedures have been detailed elsewhere [18] and are listed in Table 1. CHW home visits encompassed participant FV consumption and PA measurement and education, and used Motivational Interviewing strategies [52] to target the crucial interpersonal social support component in behavior change [53,54]. Motivational Interviewing uses collaborative interviews to internally motivate participants to set behavioral goals and to develop strategies to overcome barriers [55]. Motivational Interviewing has been effective in decreasing BMI and in increasing PA in a border Latino population similar to TSSC [55]. CHWs also invited participants to interpersonal programming and divulged information about TSSC-derived local PA and food infrastructure improvements (Table 1). CHW home visits thus served as the gateway by which participants connected with all TSSC programming facets to address multiple ecological levels that influence FV consumption and PA behaviors [56] (Table 1). This study placed a primacy on follow-up CHW home visits specifically because this was the only TSSC component reserved for participants with abnormal BMI and BP, and thus these overweight and/or hypertensive participants would derive an enhanced benefit from behavioral change interventions to augment their health trajectory [15,17].

#### 2.1.3. Tu Salud ¡Si Cuenta! Program Dissemination and Implementation

The TSSC program was initially designed for Latinos in one border city [48] to target psychosocial and environmental factors for FV consumption and PA [53,54], including cognitive and behavioral processes of change [30], to ultimately mitigate the high burden of NCD morbidity and mortality [15]. Beginning in 2014, the TSSC program was disseminated to other regional municipalities (Figure 3), using CWC model components to address multiple socio-ecological levels (Table 1) in order to promote health behavior.

### 2.2. Measures

#### 2.2.1. Sociodemographic Characteristics Measures

Sociodemographic characteristics collected from the sample included: sex (male/female), age (as a continuous variable), Latino (yes/no), health insurance (insured with government or private health insurance or uninsured), and income (above or below the federal poverty level [57]). Income level was based on two questions regarding household size and yearly household income. These variables were used to describe the reach of the program toward the intended priority population and as confounding variables in multivariable regression models of effectiveness.

#### 2.2.2. Fruit and Vegetable (FV) Consumption Measure

The validated Two-item Dietary Questionnaire for Adults was used to assess FV consumption among participants, asking “How many portions of fruit, of any sort, do you eat on a typical day?” and “How many portions of vegetables, excluding potatoes do you eat on a typical day?” [58], with pictures of locally consumed fruits and vegetables to reflect recommended portion sizes [30]. This Questionnaire has been used before with the local population in Spanish [13,59] with, in terms of concurrent validity, moderate agreement to a 24-h recall survey for fruit (kappa statistic (k) = 0.46, *p* < 0.001) and a fair agreement for vegetables (k = 0.16, *p* = 0.07) among the local Latino population [60], using standard categories of agreement based off kappa values used in prior FV validation studies [61]. Validation of this measure, however, was not the focus of this study. The threshold for meeting the current U.S. dietary recommendations was set at 5 FV servings (3 cups of vegetables, 2 cups of fruits) daily [1].

#### 2.2.3. Physical Activity (PA) Measure

The Godin-Shepherd Leisure-Time Exercise Questionnaire instrument [62,63] was used to measure the intensity, frequency, and duration of intentional PA. As utilized in prior studies [18,63], the translated instrument was translated from English into Spanish by a Spanish-fluent TSSC project member, then the Spanish form was back-translated by a different TSSC project member, and finally the back-translated form was compared to the original English survey to make revisions to ensure linguistic accuracy. This instrument has been in use before with the local population [13,18] with high test-retest reliability (r = 0.75, *p* < 0.05) and moderate agreement when compared to an accelerometer (k = 0.42, *p* < 0.05) [64], with fair agreement seen in Spanish-speaking patients (k = 0.20, *p* < 0.05) when compared to an objective pedometer measure for classifying participants as “physically active” in meeting PA guidelines or not [65]. Weekly frequency of moderate or strenuous PA was multiplied by the minutes spent on each moderate or strenuous activity to calculate each participant’s weekly metabolic equivalent adjusted minutes (MET-minutes). The metric for being “physically active” was quantified as meeting the American PA recommendations of ≥600 MET-minutes weekly [66], in tandem with prior exercise interventions [17,18,30,53,54].

#### 2.2.4. Participant Assignment to Locations

Participants were classified by the geographic location they received the TSSC program services. Almost exclusively, participants were seen within the jurisdictional boundaries of the CHW paid by that municipality or precinct. In addition to community CHWs, three CHWs were hired by the university to deliver services across multiple locations; their participants were grouped by the participant’s primary residence.

### 2.3. Statistical Analysis

All participants who qualified for and received follow-up CHW home visits formed our initial sampling frame, from which we drew our study population for analysis. Guided by the RE-AIM Framework [37] for programmatic evaluation of this dissemination and implementation study, the following approach to statistical analysis was taken to address each component of RE-AIM:

Reach: Reach was measured as the percentage of the total population living in the program locations with CHW follow-up visits to address risk factors. The total population of the municipalities/precincts (“locations”) was 718,647 based on the 2018 American Community Survey [47] in the 12 locations. This was the denominator and the total unique TSSC program participants who received CHW home visits during the study timeframe was the numerator. We also examined how many of our participants met the program’s priority population of being of Latino ethnicity and low-income, defined as being below the U.S. Federal Poverty Line [57]. For reference, the 12 program locations cumulatively have a 91.3% Latino population [42,47], with a 32.9% poverty rate among Latinos [47].

Effectiveness: This study used CHW-delivered home visits to quantify individual program exposure because eligibility for home visits was predicated on abnormal vital signs at screening, thus suggesting unhealthy blood pressure and/or weight status in these participants. Among the participants who met the inclusion criteria for analysis, including those identifying as Latino and having at least two CHW visits for change measurements, we assessed the effectiveness of TSSC program exposure to change in self-reported measurements of daily FV consumption and weekly PA. We operationalized exposure to TSSC by the quantity of CHW home visit encounters received by each participant. We classified the participants into “low exposure” (2–3 CHW visits) and “high exposure” (4–5 CHW visits) groups based on a prior pilot study [18], which found that at least 4 CHW home visits were needed to precipitate sustained PA changes beyond 6 months [18]. The program thus defined “high exposure” as participants with 4 or more longitudinal CHW home visits to enhance long-term retention of outcomes. Beyond the CHW home visits, additional program strategies that each participant voluntarily received/attended included risk factor screening, motivational text messaging, newsletters, exercise classes, weight loss support groups, and health education programming, which were included as a linear regression covariable. We compared the baseline participant characteristics between the two groups using Student’s *t*-test or its non-parametric equivalent (i.e., Wilcoxon rank sum test) for continuous variables and Chi-square test for categorical variables. The program effect was measured by comparing the changes from baseline to the last CHW visit on FV servings consumption and MET-minutes of PA between the low and high exposure groups using two separate multivariable linear regression models.

For those who did not meet daily FV consumption or weekly PA guidelines at baseline, logistic regression models assessed whether program exposure is associated with newly complying with FV or PA guidelines by the last CHW visit. Our focus, while building logistic regression models, was on understanding the association between variables and meeting FV/PA guidelines, and not on predictions or the evaluation of discrimination performance, which may require a validation process to make predictions on new data. We also assessed the program effects by each location. Potential confounding effects based on patient demographics were evaluated and adjusted accordingly in multivariable models. To account for heterogeneity among participants in their baseline measurement and duration of total follow-up, we adjusted for these variables in our models. We tested for effect modification to assess whether the program effects on behavior changes were modified by the interaction of other variables (e.g., age effect on PA based on insurance status or gender), although no significant effects were found. Underlying assumptions of linear regression and logistic regression were evaluated while developing final multivariable models. For the analysis, we used SAS (SAS Institute Inc., version 9.4, Cary, NC, USA) statistical software with a statistical significance level of 0.05.

Adoption: We examined historical records of meetings to calculate the number of municipalities/precincts offered the TSSC program for implementation (the denominator). We then examined the number of unique municipalities/precincts that implemented the program between January 2014–November 2019 as evidenced by documented CHW home visits and other TSSC programming that was performed (the numerator). This metric was thus defined as the percentage of total municipalities/precincts in the region that were offered to the program who adopted TSSC during this time frame.

Implementation: As aforementioned, the program defined “high exposure” based on a pilot study [18], as participants with 4 or more longitudinal CHW home visits to enhance long-term retention of outcomes. This number served as a marker of the fidelity of the delivery of a high dose of CHW home visits occurring in a location. A site needed to have ≥35 participants with at least 4 CHW visits to provide a sufficient sample size for stable regression analysis for location-specific effectiveness analysis, suggestive of a high delivery of the TSSC curriculum at a site. An electronic database captured CHW visits by location.

Maintenance: This was defined as the percent of locations that adopted the program and had active ongoing CHW home visits as of November 2019.

## 3. Results

### 3.1. Intervention Reach

Between January 2014 and November 2019, a total of 15,870 participants were recruited by TSSC, or about 2.21% of the entire population of the 12 partnering locations. Among participants who provided ethnicity data, 12,304 (93.25%) identified as Latino. 81.91% of those who provided both ethnicity and income data (*n* = 8225) were aligned with our priority population of Latinos with a low income.

### 3.2. Intervention Effectiveness

#### 3.2.1. Effectiveness Baseline Characteristics

Among 15,870 TSSC participants who were recruited and had an initial CHW home visit for risk factor screening, participants who were not eligible for follow-up CHW home visits due to having normal vital signs (*n* = 1557, 9.8%), or those eligible but declining to have follow-up CHW home visits (*n* = 142, 0.9%) were excluded. Those with only an initial CHW visit, despite being eligible for follow-up (*n* = 6233, 39.3%), were also excluded from the analysis, as there was no measurement of FV consumption and PA change from baseline. Non-Latinos, or those who did not disclose their ethnicity (*n* = 2252, 14.2%), were excluded from the analysis given the study’s emphasis on investigating the effect and implementation of TSSC on Latinos, specifically given the paucity of research for this population. Therefore, a total of 5686 (35.8%) participants were included in our analysis and were classified into “low exposure” (*n* = 4639) and “high exposure” (*n* = 1047) groups for analytical comparisons. The mean number of total CHW visits did not differ by sex (2.65 for females, 2.67 for males) or by 10-year age groups (range between 2.5–2.8) (not shown in Table 2). Baseline characteristics by exposure group are presented in Table 2. Median length of CHW follow-up from initial to the last CHW follow-up visit was 2.3 months for the low exposure and 6.6 months for the high exposure group (Table 2). Participants did not differ significantly in age or sex by exposure group (Table 2). The high exposure group was significantly more likely to have health insurance (49.0% vs. 40.1%, *p* = 0.0008) and less likely to be in poverty (76.6% vs. 84.2%, *p* < 0.0001). At baseline, the high exposure group had a significantly higher mean MET-minutes (306.3 vs. 166.4 MET-minutes, *p* < 0.0001), mean FV consumption (3.4 vs. 3.2 servings, *p* < 0.0001), and people who met the PA guidelines (44.0% vs. 37.9%, *p* = 0.0004), and FV guidelines (27.5% vs. 20.6%, *p* < 0.0001) (Table 2).

#### 3.2.2. Effectiveness across Municipalities on Fruit and Vegetable Consumption

Unadjusted results of mean change in FV consumption from baseline to the last CHW visit was similar to the adjusted multivariable model (Table 3). Both exposure groups had a significant increase in FV consumption from baseline to the last CHW visit, with the FV consumption increase in the high exposure group (+1.38 FV servings) being significantly higher than that of the low exposure group (+0.73 FV servings) (adjusted mean difference = +0.65 FV servings, 95% CI: 0.53–0.77; *p* < 0.0001) based on a multivariable adjusted model (Table 3). Additionally, participants with health insurance had a significantly greater FV consumption increase from baseline compared to those without insurance (adjusted mean difference = +0.39 FV servings, 95% CI: 0.29–0.49; *p* < 0.0001) (Table 3).

Among the 4411 participants who did not meet FV guidelines at baseline, 946 (21.6%) met the FV guidelines by their last CHW visit (Table 4). The high exposure group had higher odds of meeting FV consumption guidelines by the last CHW visit compared to the low exposure group (adjusted odds ratio [AOR] = 2.02, 95% CI: 1.65–2.47; *p* < 0.0001) based on the adjusted multivariate model (Table 4). Each additional program strategy received (AOR 1.30, 95% CI: 1.22–1.39; *p* < 0.0001), and having health insurance (AOR 1.24, 95% CI: 1.04–1.48; *p* = 0.0151) significantly increased the odds of meeting FV guidelines by the last CHW visit (Table 4).

#### 3.2.3. Effectiveness across Municipalities on Meeting Physical Activity Guidelines

Both exposure groups had a significantly increased mean change from baseline to the last CHW visit in PA (in MET-minutes per week) based on unadjusted and multivariable linear regression models across all 12 locations (Table 3). In the adjusted model, the PA increase from baseline to the last CHW visit in the high exposure group (+396.3 MET-minutes) was significantly higher than in the low exposure group (+210.7 MET-minutes) (adjusted mean difference = +185.6 MET-minutes, 95% CI: 105.9–265.4; *p* < 0.0001) (Table 3). A higher number of voluntary program strategies received (+147.8 MET-minutes per additional program component, 95% CI: 120.8–174.7; *p* < 0.0001), younger age (−9.3 MET-minutes per year of age increase, 95% CI: −11.3 to −7.3; *p* < 0.0001), and having health insurance (+161.7 MET-minutes, 95% CI: 95.1–288.3; *p* < 0.0001) significantly increased the PA level from baseline (Table 3).

Among the 3429 participants who did not meet PA guidelines at baseline, 1133 (33.3% of the group) met PA guidelines by their last CHW visit (Table 4). The adjusted model demonstrated that the high exposure group had significantly higher odds of meeting PA guidelines by the last CHW visit compared to the low exposure group (AOR = 1.36, 95% CI: 1.10–1.68; *p* = 0.0046) (Table 4). Increased voluntary program strategies received (AOR = 1.18, 95% CI: 1.10–1.26; *p* < 0.0001), female sex (AOR = 1.35, 95% CI: 1.12–1.61; *p* = 0.0014), and having health insurance (AOR = 1.36, 95% CI: 1.15–1.63; *p* = 0.0004) significantly increased the odds of complying with PA guidelines at the last visit. Older age (AOR = 0.98, 95% CI: 0.97–0.98; *p* < 0.0001) and being in poverty (AOR = 0.73, 95% CI: 0.59–0.91; *p* = 0.0005) significantly lowered the odds of meeting PA guidelines (Table 4).

#### 3.2.4. Effectiveness by Location on Fruit and Vegetable Intake and Physical Activity

In a location-specific analysis (Table 5), one city (City B), two county precincts, and two rural municipalities (Rural E and Rural F) did not have a sufficient sample size (*n* ≥ 35) in the high exposure group for stable regression analysis and were excluded from the location-level program effectiveness analysis. The adjusted mean changes in FV consumption and PA levels from baseline to the last CHW visit were compared between the low and high exposure groups in seven location—one urban area (City A), two towns (Town A, B), and four rural areas (Rural A, B, C, and D). The proportion of their sample in the high exposure group varied greatly across locations: 27.7% of *n* = 1251 (City A); 7.4% of *n* = 457 (Town A); 11.0% of *n* = 747 (Town B); 56.7% of *n* = 702 (Rural A); 15.4% of *n* = 540 (Rural B); 7.7% of *n* = 467 (Rural C); 6.6% of *n* = 587 (Rural D).

Based on Table 5, each location had an increase in FV consumption in both exposure groups at follow-up, with the high exposure group having significantly higher increases in FV consumption. For example, the high exposure group of Town A had significantly higher increases in FV consumption compared to the low exposure group (adjusted mean difference = +1.45 FV servings, 95% CI: 0.71–2.19; *p* < 0.0001) (Table 5). The high exposure group had higher odds of meeting FV consumption guidelines compared to the low exposure group in six locations, with four locations being statistically significant (Table 5).

For PA, the high exposure group had greater increases in PA compared to the low exposure group in five municipalities, including Town A which had an adjusted mean difference of +897.62 MET-minutes/week (95% CI: 541.05–1254.19; *p* < 0.0001) (Table 5). The high exposure group had a lower adjusted mean change of PA compared to the low exposure group in Town B and Rural B, but these findings were not significant (Table 5). Six locations saw that the high exposure group have significantly higher odds of meeting PA guidelines at the last CHW visit compared to the low exposure group, with three meeting statistical significance including in Rural C (AOR = 5.41, 95% CI: 1.62–18.08; *p* = 0.0061) (Table 5).

### 3.3. Program Adoption

During the first two-thirds of the study period (January 2014–December 2017), nine locations were offered the program and all nine adopted it (100% adoption rate). Beginning August 2017, four more locations were approached, with three accepting the program. The overall adoption rate across the region (12 adopted/13 approached) was therefore 92.3%.

### 3.4. Program Implementation

Of the 12 locations that were adopted, seven (58.3%) were implemented with fidelity in terms of providing a sufficient number of participants (≥35) sufficiently exposed to the program at a high dose of CHW home visits (≥4 home visits). The five locations that did not implement with fidelity included two locations with personnel issues (Rural E and City B), delayed administrative setbacks, and a lack of certified CHWs in the community willing to participate in providing home visits for TSSC (Precinct A, Precinct B, and Rural F). These factors delayed the program’s CHW home visit implementation and thus, insufficient numbers of participants received the required CHW home visit curriculum.

### 3.5. Program Maintenance

All but one location maintained the program (91.7%) with active CHW home visits up until November 2019. Rural E discontinued the program due to a lack of a consistently available CHW for employment despite the funding being available.

## 4. Discussion

TSSC, in leveraging the community asset of CHWs to perform home visits in a culturally tailored manner for the local Latino population [67], reached its intended priority population of low-income Latinos on the U.S.-Mexico border at the regional level. TSSC demonstrated effectiveness in significant increases in FV consumption and PA engagement behavior among TSSC participants through both low and high levels of CHW home visits, and had high adoption, moderate implementation fidelity, and high maintenance of the program across the program locations. These results echo prior CWCs that have been successfully adapted to promote FV consumption [30,68] and PA engagement [69,70] in other communities. This study contributes to the literature by examining the scaled-up dissemination of the TSSC program at a regional level for a health disparate population of low-income Latinos along the U.S.-Mexico border in Texas, who have a low uptake of healthy behaviors at baseline [13,15,18].

### 4.1. Reach

Overall, we enrolled 2.2% of the total population of the participating locations for an initial CHW home visit and 81.91% of our sample was our priority population of low-income Latinos. This indicates that the CWC model can effectively enroll the priority population for which programming services were designed, even in a resource-scarce region. This study contrasts with the concern from prior community interventions on the rare reporting of priority population reach and that minorities are less likely to enroll in and benefit from community health interventions [71,72]. Increasing public awareness via expansion of a local media campaign highlighting TSSC [48], particularly to regions farther away from the original program epicenter as the program expands, may improve the willingness of unenrolled participants to engage in TSSC.

### 4.2. Effectiveness

TSSC is the first CWC intervention in the literature that examined the program effects on both FV consumption and PA engagement. We found a significant improvement in FV consumption and PA aggregately across locations, with a greater improvement in the high compared to the low exposure group, but with significant increases, even with 1–2 CHW follow-up visits. Our findings parallel CHW-based Latino community interventions that improved behavioral outcomes [9,13,17,18,30]. TSSC’s FV consumption improvements corroborate with prior CWCs designed to decrease the intake of high-fat dairy in resource-poor American communities [73,74], saturated fat in American [75] and Dutch adults [76], and red meat in Iranian adults [77]. Having health insurance was independently associated with mean increases in FV and PA behaviors in TSSC. Possessing health insurance likely signified a higher socioeconomic status and legal residency/citizenship status, which enhances the ability to engage in the facets of this longitudinal program [78]. Having insurance can presumably suggest a more routine access to healthcare, where healthcare providers may also provide supplemental encouragement for the participant to engage in healthy behaviors. Past research suggests that multimodal CWC models are effective for dissemination and implementation in communities with high needs [79]; our study results corroborated this finding particularly, as increasing exposure to TSSC program modalities, beyond just CHW home visits, were also independently associated with increased FV consumption and PA engagement. CHW Motivational Interviewing training is noted to be rarely done in practice [80] but has been effective in delivering healthy dietary and PA behavior change to border Latinos [55]. Future interventions in Latino communities can incorporate systematic training of Motivational Interviewing for CHWs to empower them to provide more effective health services given their critical liaison role of providing services for low-income Latinos [55,80].

In a location-specific analysis, the effect size difference by location is wide, differs by outcome, and is partially explained by individual and organizational capacity factors. The community CHWs differed individually in their professional backgrounds and approach to follow-up visits. Each municipality had varying levels of priority placed on this program as evidenced by the varied time spent in an administrative setup by different municipalities before initiating CHW home visits. As the CHWs balanced municipal and program demands, their emphasis on providing CHW home visits may have diverged. Future research should examine factors associated with RE-AIM’s Implementation and Maintenance dimensions [37] to comprehend program effect variability by location.

In line with our study, CWCs, through targeting multiple levels of the ecological environment, have shown significant effects in improving PA across different adult populations globally [21,24,81], and in improving FV consumption in one study on African-Americans [68]. A Japanese study, performed on older adults aged 40–79, took longer to yield significant behavioral change (5 years) with a more modest effect size than other CWC PA studies [69,82]. Future CWCs should consider the age demographics of their priority population, as this can influence their program effect.

Moreover, no prior CWC had published PA effects by socioeconomic status or other markers of equity [26]. This study thus presented a CWC intervention FV and PA effects in a resource-limited American Latino population by socioeconomic status, including poverty rate and insurance status. Future CWCs, regardless of setting, should assess the programmatic effects by socioeconomic status indicators, as CWCs can be used as a vehicle to tackle disparities in preventive health behaviors to promote health equity, particularly for low-resource and disenfranchised populations throughout the world [83]. Since CWCs are effective for behavioral change [26], CWCs should strive to focus on low-resource and ethnic minority populations given their disproportionate lack of engagement in health behaviors and the burden of NCDs in both low-income countries [84,85] and in high-income countries [15,25,86,87]. For TSSC’s bicultural local population [41], future analysis should also examine TSSC’s effect by acculturation level, which is another marker of equity that serves as a determinant for behavior engagement [87], in order to advance health equity.

### 4.3. Adoption

The TSSC program was adopted by 92.3% of locations to which it was promoted. Our study reflects high motivation to adopt the program in a high need region. Enthusiasm for adoption was likely influenced by the presence of grant funding to cover annual program costs. Over time, the local municipalities were able to leverage this core funding into additional programming and infrastructure improvements, which built enthusiasm. Generally, past research has indicated that community-based interventions are not adopted because health issues are not perceived by key stakeholders and municipal decision-makers as relevant [88,89,90]. Notably, the TSSC program builds upon prior local adoption of this program [9,17,30], and leverages those successes to create relevance, and handle perceived barriers and challenges to facilitate collaboration and program adoption. Indeed, the continued successes of the program will be highlighted to other municipalities in order to continue to generate a high level of interest for TSSC.

### 4.4. Implementation

The goal of TSSC was to have as many participants have longitudinal CHW home visit follow-ups to address the socioecological factors that preclude FV and PA behavioral uptake in the priority population. There was some lack of fidelity in program implementation as 39% of those eligible for follow-up CHW home visits based on abnormal BP and/or BMI did not engage in the home visits or any other TSSC component. Moreover, seven of twelve locations (58.3%) reached a minimum threshold of implementation where a sufficient dose of monthly CHW home visits was provided. Incentivizing participant engagement in follow-up CHW home visits could also be considered in the future.

In general, locations that adopted TSSC with fidelity generally expeditiously identified community CHWs with experience and capacity to implement the home visits and program components. Several sites had difficulties in finding qualified CHWs, which curtailed the number of follow-up visits performed at those locations. These challenges were seen in prior Texas Latino CHW-based interventions in which their implementation and effect success was mitigated by CHWs’ supply shortages, their lack of continuing training opportunities, and their heavy workload [67,91,92]. TSSC relied heavily on existing community CHWs who, being paid by their municipality, must balance project aims with municipal priorities. This factor influenced their proactiveness in providing longitudinal home visits. To address this in the future, TSSC can seek to provide financial and technical support for community members to receive or renew their CHW credentialing to become TSSC CHWs entirely devoted to the program, who can travel to different sites to perform longitudinal CHW home visits based on need. Furthermore, implementation fidelity can be enhanced by personnel training, as well as monthly monitoring of the dosage of program components delivered, and adherence to program component protocols at each location [93]. With this iteration of the program, we began to host monthly CHW training sessions to enhance programming consistency, along with location-specific annual implementation evaluations with location-specific CHWs and community leaders. The documentation of these monthly training sessions will become increasingly vital to the sustained dissemination of the program. As the program continues to mature and more sites are able to provide a high level of exposure to CHW home visits, the detection of significant results will improve.

### 4.5. Maintenance

Maintenance was high, but like adoption, it was influenced by the presence of an ongoing grant funding to support the intervention activities. As the program has future expansion planned, more knowledge about maintenance will come over time, as seen in the original CWC program implementation in Finland, which has been maintained for 40 years [34,35]. TSSC collaborated with a Community Advisory Board to adapt the CWC to the local population and refine it [48]; this collaborative community action of incorporating local stakeholders to instigate their vested ownership over TSSC’s program goals influenced municipal authorities to maintain this program.

### 4.6. Limitations

The limitations to our study include a lack of control communities. However, several previous studies with control groups on the TSSC program have shown evidence of efficacy [17,18,30], and this study focused on the scaled-up dissemination and implementation of the program across an entire health disparate region. This study advances the CWC literature by demonstrating how the leveraging of community assets in a culturally tailored manner can address behaviors such as PA at a regional level, advancing the knowledge of the reach, effectiveness, adoption, implementation, and maintenance of this model in a real-world scenario in a Latino population.

Another limitation is the differential sample size between the low and high exposure groups. The unequal sample size between these groups may have weakened the statistical power and resulted in the exclusion of five of the twelve implementing locations from the analysis. As more participants are followed for a longer timeframe, the program effect and maintenance can be examined using longitudinal analysis. The outcome variables were measured through self-reported measures subject to a intervention-induced social desirability bias that can cause over-reporting of the participant-reported measures in FV and PA with resultant differential misclassification of these measures and over-estimation of the true effect size of behavior change in our study [94]. The two-item dietary questionnaire and the Godin-Shepherd Leisure-Time Exercise Questionnaire had only a moderate validity for American Latinos [60,64], which can impact the interpretation of the measured FV intake and PA levels in this study. The lower validity with these behavioral measures is related to its self-reported nature, where respondents can over-report their behavioral activity [95]. However, the use of objective measures tends to be costly for large populations [54], burdensome to participants, and logistically difficult [69]. Future research could consider a subsample cohort that is followed with more objective measures, such as the Automated Self-Assessment (ASA) for FV [96] and Accelerometers for PA [54].

Moreover, FV consumption and PA engagement behavioral changes are not just due to the CHW home visits by itself, but the delivery of the entire multimodal components of TSSC. The synergy of all these different facets, each impacting various socioecological levels of influence, is likely what ultimately yields FV consumption and PA behavioral changes. Our analysis quantified program exposure based on CHW home visits given its primacy in the TSSC curriculum with a regression covariate established for program components received. However, future programmatic enhancements can include enhancing the tracking of participant engagement with each component of TSSC to better quantify the isolated effect of each individual program component on behavior.

### 4.7. Strengths

This study’s strength rests with its large sample of 5686 predominately low-income Latino adults along the U.S.-Mexico border who at baseline have disproportionately low uptake of FV consumption and PA engagement [8,9,13,15,45], and face systemic inaccessibility to health promotion services [17,18,40,44]. Accordingly, this study detailed the real-world dissemination and implementation of the community-based TSSC behavior promotion program to this understudied, underserved population.

Other CWCs have examined effectiveness, but rarely have dissemination and implementation been studied. We discovered novel evidence that the broad dissemination and implementation of a CWC in a low-resource region is related to enhanced uptake of FV and PA behaviors in thousands of Latinos in diverse, real-world settings regionally. However, we went further and examined facilitating and inhibiting factors that influenced our scaled-up dissemination and implementation of TSSC to inform and improve next generation implementation efforts of TSSC and CWCs in general. It would be of interest to frame CWCs as a modality for advancing health equity, and thus the intersectionality of socioeconomic status and health behavior uptake in community interventions and the sustainability of FV and PA behavioral changes should be explored. Public health initiatives will continue to benefit from the explication of implementation efforts based on the RE-AIM framework because practitioners generate insight into how to practically disseminate and implement evidence-based interventions into tangible community settings.

## 5. Conclusions

This Effectiveness-Implementation Hybrid Type I study used the RE-AIM Framework [37] to evaluate the dissemination and implementation of the TSSC program, an adapted CWC, across 12 locations along the U.S.-Mexico border in Texas. To address local needs, TSSC mobilized CHWs in culturally tailored home visits to deliver motivational interviewing to instigate intrinsic motivation for FV consumption and PA, and augment interpersonal social support. Overall, the TSSC program, through its multimodal impacts on the local ecological environment, affirmed our hypothesis in being able to reach its intended low-income Latino population of Mexican descent who at baseline have a low uptake of health behaviors including PA [13]. The study revealed that, in concordance with our hypothesis, participants with 3–4 follow-up CHW home visits had an increased degree of statistically significant increases in FV consumption and PA compared to those with 1–2 follow-up CHW home visits. No prior CWC had published effects by socioeconomic status or other markers of equity [26]. This study demonstrated that having insurance had a significant independent effect on FV and PA increases, and that being in poverty led to significant independent decreases of PA.

The study implementation revealed high adoption, moderate fidelity, and high maintenance across a variety of settings ranging from major urban to rural communities. CHW delivery and implementation funding positively influenced reach, effectiveness, adoption, and maintenance. At the organizational level, a lack of qualified CHWs and differential proactiveness to conduct longitudinal home visits among CHWs negatively impacted the fidelity of implementation in some locations. Evidence-based health behavior promotion interventions in Latino communities rarely analyze the reach, adoption, implementation, or sustainability of interventions [53]; accordingly, this study adds evidence about the implementation of the scaled-up dissemination of an adapted CWC model, TSSC, to address multiple socioecological factors to induce positive behavior change in low-income Latinos with abnormal BMI and/or BP as a non-clinical marker of having the NCDs of obesity and/or hypertension. Indeed, this study demonstrates that in a resource-poor region, a health promotion intervention tailored to the assets and needs of the community can generate health behavior change among thousands. These findings will be used for iterative enhancement of TSSC’s continued dissemination and implementation to target the ecological factors that influence health behaviors among Latinos in a health disparate region to advance health equity.

## Figures and Tables

**Figure 1 ijerph-19-04514-f001:**
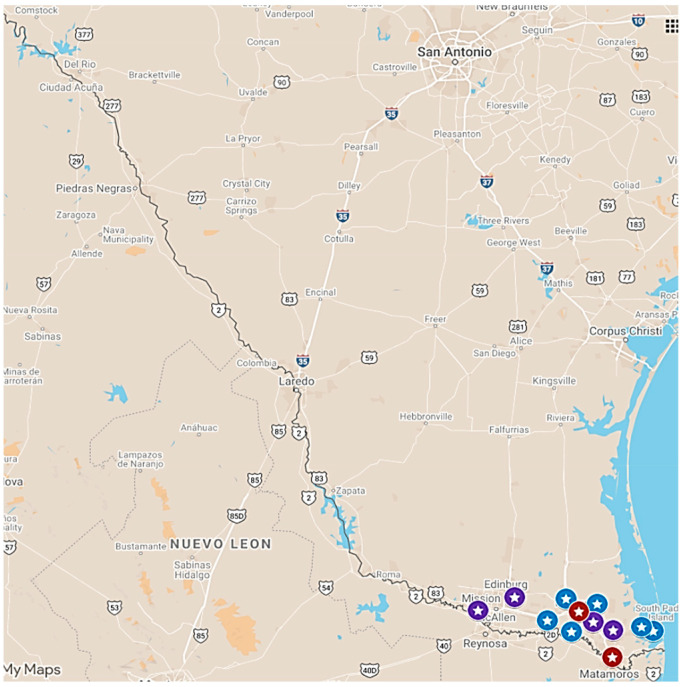
Map of Participating TSSC Implementing Locations. Google map showing the location of the 12 municipalities in the TSSC study along the U.S.-Mexico border. Red labels are “Cities” with population sizes ranging from 75,000–200,000. Purple labels are “Towns” with a population sizes from 10,000–70,000. Blue labels are “Rural” areas, each with a population of <5000. The straight-line distance between the farthest west (purple star) and east (blue star) locations is about 80 miles (130 km).

**Figure 2 ijerph-19-04514-f002:**
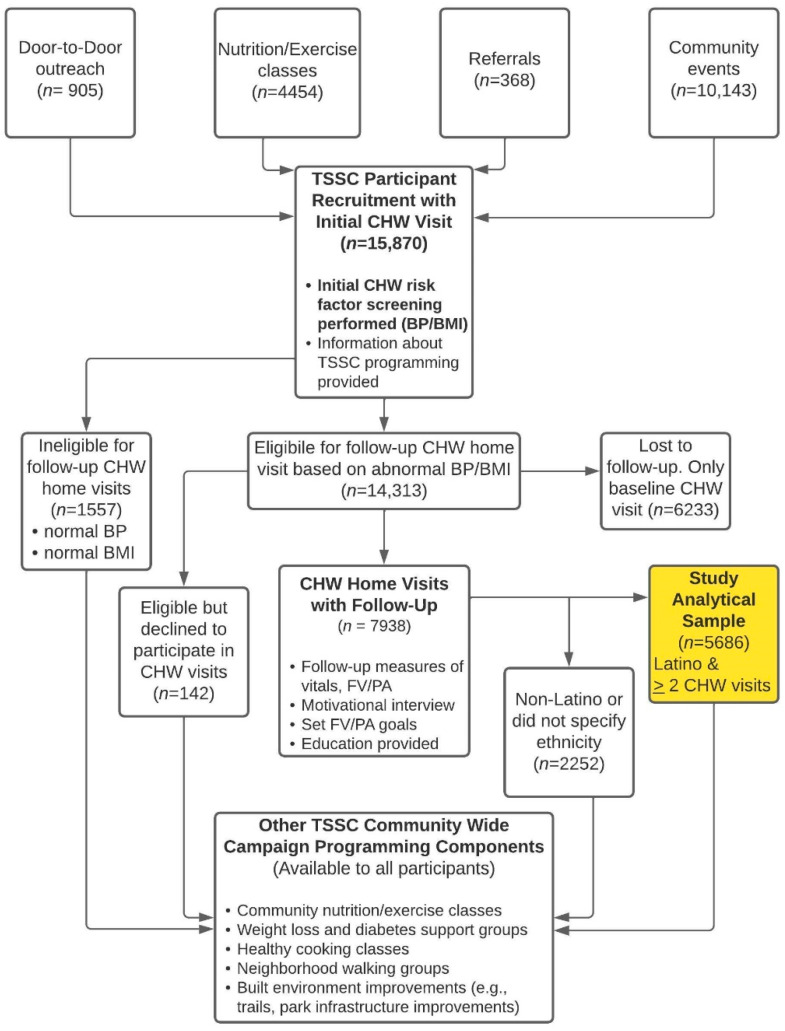
Flowchart of TSSC Components. This flowchart visually portrays TSSC’s participant recruitment and the enrollment process for participants to receive follow-up CHW home visits. This study focused on Latino TSSC participants with follow-up CHW home visits (in yellow); however, eligible and willing participants of any ethnicity received follow-up visits. BMI, body mass index; BP, blood pressure; CHW, community health worker; FV/PA, fruit and vegetable intake and physical activity engagement.

**Figure 3 ijerph-19-04514-f003:**
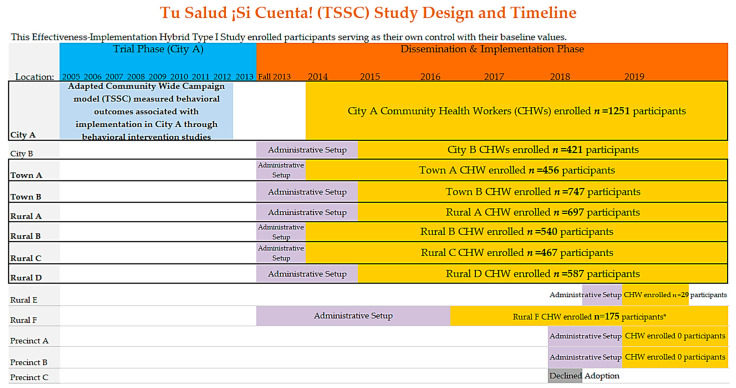
TSSC Dissemination and Implementation Timeline. Sample size only represents participants at each site that completed a minimum of 2 community health worker (CHW) home visits by study conclusion (11/2019). The width of colored bars in each row indicate the length of time the TSSC intervention was implemented at a location. Bolded sites with thick borders indicate a sufficient sample size (*n* ≥ 35) in the “High Dose Exposure” group (≥4 CHW Home Visits) for our location-specific analysis. The administrative setup indicates that the site agreed to adopt TSSC, contractual agreements were finalized, administrative arrangements were provided, and personnel were brought onboard.

**Table 1 ijerph-19-04514-t001:** Tu Salud ¡Si Cuenta! Community-Wide Campaign Program Description.

Community-Wide Campaign Components	Target Socioecological Level [56]	Operationalization of Component to Promote Fruit and Vegetable (FV) Consumption and Physical Activity (PA)
Risk Factor Screening	Individual	Provision of free and accessible health risk screening in home and community locations for: (1) Blood Pressure (2) BMI (3) Referrals for glucose/HbA1c (4) Referral to clinical and social services
Mass and Social Media	Individual, Interpersonal	Delivery of regularly scheduled health promoting messages tailored to local culture and language (social media, radio, TV, print, text messaging), including sharing stories of weight loss and new PA classes in the community, using behavioral journalism featuring local role models.
Social Support and Tailored Health Education	Individual, Interpersonal, Organizational, Community	Community-based groups tailored to local culture, language and identified health needs of community members including: (1) Follow-up community health worker home visits to set FV and PA goals and progress (2) Individualized motivational interviewing to increase intrinsic motivation to change lifestyle behaviors based on personal values and goals (3) Motivational text messaging (4) Exercise and nutrition groups (5) Weight loss and diabetes support groups (6) Community health education programming
Environmental Changes	Community, Policy	Physical improvements identified and implemented by the locations including: (1) Availability and access to produce through community gardens and farmers markets (2) Activation of public spaces through free PA classes (3) Community connectedness projects like sidewalks and trails (4) Safety and convenience projects like outdoor hydration and lighting
Policy Improvements	Policy	Regulation improvements by locations including: (1) Complete street ordinances and healthy vending policies (2) Improved school nutrition policies (3) Active transportation policies

**Table 2 ijerph-19-04514-t002:** Baseline characteristics of Latino participants enrolled in the TSSC Program (*n* = 5686).

Variable	Low Exposure (*n* = 4639)	High Exposure (*n* = 1047)	*p*-Value ^a^
**Age**, years, mean (SD)	46.87 (14.06)	48.62 (14.13)	0.8298
**Female**, *n* (%)	3644 (77.49%)	819 (78.22%)	0.8156
**Have insurance,***n* (%)	**1862 (40.14%)**	**512 (48.95%)**	**<0.0001**
**Length of follow-up,** months, median (IQR)	**2.3** (1.1, 5.1)	**6.6** (4.9, 9.3)	**<0.0001**
**Number of program strategies received**^b^, median (IQR), [min, max]	**2 (1, 2)** [1,7]	**2 (1, 3)** [1,7]	**0.0035**
**Below federal poverty level,***n* (%)	**3312 (84.22%)**	**706 (76.58%)**	**<0.0001**
**Meeting PA guideline at Baseline,***n* (%)	**1747 (37.89%)**	**443 (43.95%)**	**0.0004**
**Meeting Fruit & Vegetable guideline at Baseline**, *n* (%)	**952 (20.60%)**	**281 (27.50%)**	**<0.0001**
**Baseline MET-minutes,** mean (SD)	**166.35 (215.57)**	**306.33 (266.88)**	**<0.0001**
**Baseline FV consumption servings/day,** mean (SD)	**3.15 (2.06)**	**3.38 (2.24)**	**0.0015**

Abbreviations: SD, standard deviation; IQR, interquartile range; MET-minutes, metabolic equivalent minutes of physical activity. ^a^ Student’s *t*-test or its non-parametric equivalent (i.e., Wilcoxon rank sum test) for continuous variables and Chi-square test for categorical variables were used. ^b^ Number of program strategies received per participant has a range from 1 to 7 and includes CHW home visits (participation of which was mandatory for participants included in this study), risk factor screening, motivational text messaging, newsletter, exercise classes, weight loss support groups, and health education programming. **Bold** numbers indicate statistically significant results (*p* < 0.05).

**Table 3 ijerph-19-04514-t003:** TSSC program effect on change in FV and PA MET-minutes from baseline to last visit based on linear regression analysis (*n* = 5686).

Independent Variables Column:	Change in Total FV Consumption from Baseline to the Last CHW Visit				Change in Total PA MET-Minutes from Baseline to the Last CHW Visit			
	Unadjusted a		Adjusted b		Unadjusted c		Adjusted d	
	Mean Difference in Change (95% CI)	*p*-Value	Mean Difference in Change (95% CI)	*p*-Value	Mean Difference in Change (95% CI)	*p*-Value	Mean Difference in Change (95% CI)	*p*-Value
**Exposure** high vs. low	**0.67** **(0.56, 0.79)**	**<0.0001**	**0.65** **(0.53, 0.77)**	**<0.0001**	**168.55** **(87.90, 249.21)**	**<0.0001**	**185.62** **(105.87, 265.37)**	**<0.0001**
** *Estimated Mean change (95% CI)* ** *from baseline to the last CHW visit in each group*	*Low: 0.63 (0.58, 0.67)* *High: 1.30 (1.20, 1.40)*		*Low: 0.73 (0.65, 0.80)* *High: 1.38 (1.26, 1.50)*		*Low: 138.14*		*Low: 210.67*	
					*(106.45, 169.83)*		*(162.09, 259.26)*	
					*High: 306.69*		*High: 396.29*	
					*(234.67, 378.72)*		*(316.80, 475.79)*	
# **program strategies received**			**0.16** **(0.12, 0.20)**	**<0.0001**			**147.77** **(120.83, 174.72)**	**<0.0001**
**Age**, year			0.002 (−0.001, 0.005)	0.1663			**−9.26** **(−11.28, −7.25)**	**<0.0001**
**Sex** female vs. male			−0.06 (−0.16, 0.04)	0.2438			−30.21 (−98.24, 37.82)	0.3840
**Have insurance** yes vs. no			**0.39** **(0.29, 0.49)**	**<0.0001**			**161.72** **(95.14, 228.30)**	**<0.0001**
**Poverty status** below vs. above FPL			0.05 (−0.07, 0.18)	0.3789			−66.04 (−147.10, −15.02)	0.1103

Dependent variable: a and b, change in total FV consumption; c and d, change in total PA MET-minutes. Abbreviations: CHW, community health worker; FV, fruit and vegetable; MET, metabolic equivalents; PA, physical activity. High program exposure defined as participants having 4–5 CHW visits (3–4 follow-up visits). Low program exposure defined as participants having 2–3 CHW visits (1–2 follow-up visits). The “estimated mean change” row indicates the specific changes in FV and PA behavior within each exposure group and is *italicized* to demonstrate that this row is detailing at the individual group level the mean change described in the “exposure” row. The gray background color indicates for the unadjusted model, the variables in the corresponding rows did not adjust for those variables and thus no results are available to present. **Bold numbers** indicate statistically significant results (*p* < 0.05).

**Table 4 ijerph-19-04514-t004:** TSSC program effect on meeting FV and PA guidelines based on logistic regression analysis (*n* = 5686).

Independent Variables Column:	Meeting FV Guideline by Last CHW Visit (*n* = 4411)				Meeting PA Guideline by Last CHW Visit (*n* = 3429)			
	Unadjusted a		Adjusted b		Unadjusted c		Adjusted d	
	Odds Ratio (95% CI)	*p*-Value	Odds Ratio (95% CI)	*p*-Value	Odds Ratio (95% CI)	*p*-value	Odds Ratio (95% CI)	*p*-Value
**Exposure** high vs. low	**2.00** **(1.64, 2.43)**	**<0.0001**	**2.02** **(1.65, 2.47)**	**<0.0001**	**1.24** **(1.10, 1.54)**	**0.0393**	**1.36** **(1.10, 1.68)**	**0.0046**
# **program strategies received**			**1.30** **(1.22, 1.39)**	**<0.0001**			**1.18** **(1.10, 1.26)**	**<0.0001**
**Age**, year			1.00 (1.00, 1.01)	0.2430			**0.98** **(0.97, 0.98)**	**<0.0001**
**Sex** female vs. male			1.18 (0.98, 1.42)	0.0737			**1.35** **(1.12, 1.61)**	**0.0014**
**Have insurance** yes vs. no			**1.24** **(1.04, 1.48)**	**0.0151**			**1.36** **(1.15, 1.63)**	**0.0004**
**Poverty status** below vs. above FPL			1.02 (0.82, 1.26)	0.8951			**0.73** **(0.59, 0.91)**	**0.0005**

**Dependent variable:** a and b, meeting FV guideline (yes/no); c and d, meeting PA guideline (yes/no). Abbreviations: CHW, community health worker; FV, fruit and vegetable; FPL, federal poverty line; PA, physical activity. High program exposure defined as participants having 4–5 CHW visits (3–4 follow-up visits). Low program exposure defined as participants having 2–3 CHW visits (1–2 follow-up visits). The gray background color indicates for the unadjusted model, the variables in the corresponding rows did not adjust for those variables and thus no results are available to present. **Bold numbers** indicate statistically significant results (*p* < 0.05). **Test Statistics for the main effect (high vs. low exposure)** a: 42.86; b: 40.22; c: 1.41; d: 3.59.

**Table 5 ijerph-19-04514-t005:** TSSC program effect on change in FV, PA MET-minutes and meeting FV, PA guideline by location (*n* = 4751).

Change in Total FV Consumption from Baseline to the Last CHW Visit a							
Independent Variables Column:	City A (*n* = 1251)	Town A (*n* = 457)	Town B (*n* = 747)	Rural A (*n* = 702)	Rural B (*n* = 540)	Rural C (*n* = 467)	Rural D (*n* = 587)
	Adjusted Mean Difference in Change and 95% Confidence Interval with *p*-Value						
**Exposure** high vs. low	**0.63** **(0.37, 0.88); *p* < 0.0001**	**1.45** **(0.71, 2.19); *p* < 0.0001**	**0.62** **(0.37, 0.88); *p* < 0.0001**	**0.63** **(0.44, 0.83); *p* < 0.0001**	**0.71** **(0.17, 1.24); *p* = 0.0097**	**0.74** **(0.24, 1.26); *p* = 0.0041**	**0.44** **(0.06, 0.82); *p* = 0.0242**
** *Estimated Mean change (95% CI)* ** *from baseline to the last CHW visit in each group*	** *High: 1.28 (1.03, 1.53)* **	** *High: 2.13 (1.31, 2.95)* **	** *High: 1.15 (0.94, 1.36)* **	** *High: 1.37 (1.23, 1.50)* **	** *High: 1.62 (1.06, 2.18)* **	** *High: 2.20 (1.50, 2.91)* **	** *High: 0.99 (0.39, 1.59)* **
	** *Low: 0.66 (0.48, 0.83)* **	** *Low: 0.68 (0.27, 1.09)* **	** *Low: 0.53 (0.40, 0.65)* **	** *Low: 0.73 (0.58, 0.89)* **	** *Low: 0.91 (0.61, 1.22)* **	** *Low: 1.45 (1.01, 1.90)* **	** *Low: 0.55 (0.03, 1.08)* **
**Meeting FV guideline at the last CHW visit b**							
	**City A (*n* = 708)**	**Town A (*n* = 304)**	**Town B (*n* = 422)**	**Rural A (*n* = 406)**	**Rural B (*n* = 307)**	**Rural C (*n* = 322)**	**Rural D (*n* = 432)**
	Adjusted Odds Ratio and 95% confidence interval with *p*-value						
**Exposure** high vs. low	**2.04** **(1.41, 2.96); *p* = 0.0002**	1.97 (0.77, 5.06); *p* = 0.1567	0.84 (0.38, 1.84); *p* = 0.6636	**3.98** **(2.19, 7.24); *p* < 0.0001**	**2.40** **(1.20, 4.82); *p* = 0.0135**	**4.10** **(1.36, 12.45); *p* = 0.0125**	2.73 (0.94, 7.94); *p* = 0.0652
**Change in Total PA MET-minutes from baseline to the last CHW visit c**							
	**City A (*n* = 1251)**	**Town A (*n* = 456)**	**Town B (*n* = 747)**	**Rural A (*n* = 702)**	**Rural B (*n* = 540)**	**Rural C (*n* = 467)**	**Rural D (*n* = 587)**
	Adjusted mean difference in change and 95% confidence interval with *p*-value						
**Exposure** high vs. low	38.55 (−102.20, 179.30); *p* = 0.5911	**897.62 (541.05, 1254.19); *p* < 0.0001**	−171.11 (−345.97, 3.74); *p* = 0.0551	**220.38** **(65.05, 375.70); *p* = 0.0055**	−147.53 (−345.83, 50.76); *p* = 0.1445	290.94 (−55.44, 637.33); *p* = 0.0995	189.59 (−122.59, 501.65); *p* = 0.2333
** *Estimated Mean change (95% CI)* ** *from baseline to the last CHW visit in each group*	*High: 34.64* *(−104.02, 173.31)*	** *High: 1409.83 (1031.81, 1787.85)* **	*High: −122.60 (−308.31, 63.11)*	** *High: 798.06 (692.83, 903.28)* **	*High: 8.28* *(−199.60, 216.16)*	** *High: 586.94 (107.68, 1066.19)* **	*High: −24.24* *(−456.66, 408.15)*
	*Low: −3.91 (−103.26, 95.45)*	** *Low: 512.21 (350.98, 673.45)* **	*Low: 48.52 (−42.99, 140.02)*	** *Low: 577.68 (458.00, 697.35)* **	** *Low: 155.81 (42.09, 269.54)* **	*Low: 295.99 (−5.86, 597.84)*	*Low: −213.82 (−706.87, 278.22)*
**Meeting PA guideline at the last CHW visit d**							
	**City A (*n* = 1007)**	**Town A (*n* = 361)**	**Town B (*n* = 608)**	**Rural A (*n* = 445)**	**Rural B (*n* = 375)**	**Rural C (*n* = 375)**	**Rural D (*n* = 511)**
	Adjusted Odds Ratio and 95% confidence interval with *p*-value						
**Exposure** high vs. low	1.15 (0.73, 1.82); *p* = 0.5382	**3.84** **(1.05, 13.95); *p* = 0.0414**	1.38 (0.56, 3.39); *p* = 0.4903	**2.42** **(1.35, 4.32); *p* = 0.0029**	1.00 (0.47, 2.10); *p* = 0.9923	**5.41** **(1.62, 18.08); *p* = 0.0061**	1.14 (0.42, 3.08); *p* = 0.7933

**Dependent variable**: a, change in total FV consumption; b, meeting FV guideline (yes/no); c, change in total PA MET-minutes; d, meeting PA guideline (yes/no). Abbreviations: CHW, community health worker; FV, fruit and vegetable; MET, metabolic equivalents; PA, physical activity; TSSC, *Tu Salud* ¡*Si Cuenta*! program. High program exposure was defined as participants having 4–5 CHW visits (3–4 follow-up visits). Low program exposure defined as participants having 2–3 CHW visits (1–2 follow-up visits). Linear regression models were performed for the mean difference in the change of FV and PA in each location. Logistic regression models were performed on all individuals who did not meet FV or PA guidelines at baseline in each location. The “estimated mean change” rows indicates the specific changes in FV and PA behavior within each exposure group and is *italicized* to demonstrate that these rows are detailing at the individual group level the mean change described in the “exposure” rows immediately above. **Bold numbers** indicate statistically significant results (*p* < 0.05). All estimates are based on multivariable models after controlling for the number of program strategies received, the duration of follow-up, age, gender, insurance status and poverty status. Detailed information for the independent effect of each of the aforementioned adjusted factors are available in Appendix A.

## Data Availability

The data presented in this study are available on request from the corresponding author. The data are not publicly available due to privacy restrictions to protect the anonymity of partnering locations, CHWs, and participants involved in the study.

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
