# Peer review of "Evaluating the Dissemination and Implementation of a Community Health Worker-Based Community Wide Campaign to Improve Fruit and Vegetable Intake and Physical Activity among Latinos along the U.S.-Mexico Border"

_ijerph, 2022, doi:10.3390/ijerph19084514_

Round 1

Reviewer 1 Report

I am grateful for the opportunity to review the interesting topic „Evaluating the Dissemination and Implementation of a Community Health Worker-Based Community Wide Campaign to Improve Fruit & Vegetable Intake and Physical Activity among Latinos along the U.S.-Mexico Border“.

The manuscript is interesting. However, I have observations:

It is necessary to specify the type of study carried out.

It is necessary to identify the design of the study and to specify the purpose.

Lines 121-128: The methods set out in this paragraph are an overview of the literature.

The Methods lack a calculation of the representative sample size. To my opinion, this section also contains a lot of unnecessary information. It seems appropriate to illustrate the design of the study, the enrollment of subjects and the study procedures carried out in an infographic manner. A description of the statistical analysis of the data should be refined and specified (which statistical tests have been used, systematically indicate dependent and independent variables, etc.).

In the Results, the citation of literature sources must be discarded. The tables were not prepared correctly. The recommendations of the IJERPH must be followed. Logic models must be followed by criteria of goodness-of-fit (Nagelkerke R2 statistic).

Confounders must also be presented as adjusted factors after each multivariate analysis (MVA) displayed in tables. The logic models must include Wald test statistic.

The consclusions must confirm or deny the hypotheses of the investigation and must therefore be revised.

It is doubtful whether the design of the study allowed authors to draw conclusions about the cause and effect, such as: „The study revealed a dose-dependent relationship between increased program exposure and degree of statistically significant increases in FV consumption and PA“.

Authors should check whether is it recommended to write Abbreviations after References.

The strong side of the article relates to a large sample size. Nevertheless, major structural and systemic corrections of the manuscript are must be done.

Best Regards

Reviewer 2 Report

The manuscript entitled “Evaluating the Dissemination and Implementation of a Community Health Worker-Based Community Wide Campaign to Improve Fruit & Vegetable Intake and Physical Activity among Latinos along the U.S.-Mexico Border” presents interesting issue, however some corrections are needed

The paper is well written and gives a good overview of the issue, however some small typos occurs. Moreover, some questions and concerns arise after reading the manuscript:

  • ‘All enrolled TSSC participants went through an initial CHW home visit’ – how this home/ participants were selected? More detailed information about recruitment procedure should be presented.
  • Fruit and Vegetable (FV) Consumption Measure ‘Two-item Dietary Questionnaire for Adults’ – such approaches have some pros and cons. In case of disadvantages please provide proper information in limitation section (at the end of the discussion section).
  • More information is needed about the validity and reliability of each measure. Additionally, any limitations in reliability and validity need to be addressed in the discussion.
  • The Godin-Shepherd Leisure-Time Exercise Questionnaire instrument - What is the original language of the questionnaire? Was the questionnaire translated? Who did so? Any validation of the translated questionnaire? Please, provide information about the validity and reliability.
  • Section ‘Statistical Analysis’ is untypical – please reorganized this section.
  • ‘informed consent was obtained from all subjects involved in the study.’ – It was written or oral informed consent?
  • Moreover some international context is missing (please add some comparison with international studies (what other countries could learn from this study?)

Reviewer 3 Report

Evaluating the Dissemination and Implementation of a Com-2 munity Health Worker-Based Community Wide Campaign to 3 Improve Fruit & Vegetable Intake and Physical Activity among 4 Latinos along the U.S.-Mexico Border

Introduction

  1. The introduction is too long. I suggest the author scale it down and only focus of the previous findings and the gab for evaluates the real-world dissemination of the evidence-based TSSC program to 12 municipalities along the U.S.-Mexico border from Jan-109 uary 2014 - November 2019

-           

  1. Line 103, the research questions section the authors has provided notes instead of scaling down what will be done in the current study

-           

-          Methodology

Line 153-1560 “community 153 CHWs, paid by the local municipality they live in and possessing varied educational back- grounds, to deliver home visits to all CHW participants while receiving monthly training 155 from TSSC research personnel.”. 1.Specify the educational background

  1. Specify the type of training they received

Line 158 -159 “All enrolled TSSC participants went through an initial CHW home visit, where base- health was assessed,” It will be of interest to some readers to know the base line health that was assessed and how it was assessed.

-          It would be of interest to some readers know the level of PA and FV consumption before the campaign started and how they were assessed.?

-          Motivational Interviewing strategies was used in the study. However the reliability and validity of information obtained using these strategy was not adequately addressed.

-          Line 168- 169 “An initial baseline CHW home visit was mandatory for all participants, but participants could choose whether to engage further with follow-up.” The total number of CHW at baseline and the follow up by gender and age will interest some readers together with the participating individual members of the population

-          The selection of the population participants or individual within the selected population not well articulated in the study

-          Statistical analysis: Software used is not shown, and the individual tests used to measure the different variables in line with the aim of the study not well articulated.

-           

Results

The cut off points for high and low exposure will interest other readers

It is not clear how Program Adoption was measured in this study

It is not clear as to the rationale for selecting the sufficient number of participants (≥35) in the implementation strategies

Round 2

Reviewer 1 Report

Thank you for the opportunity to review an interesting topic „Evaluating the Dissemination and Implementation of a Community Health Worker-Based Community Wide Campaign to Improve Fruit & Vegetable Intake and Physical Activity among Latinos along the U.S.-Mexico Border“

The authors partly responded to my comments.

Nonetheless, the authors point out that they carried out a quasi-experiment. When reading the whole methodology, the specific scheme of the experiment in question remains unclear.

A quasi-experiment is a type of research design that attempts to establish a cause-and-effect relationship. The main difference with a true experiment is that the groups are not randomly assigned.

Lines 280, 292, 293: does Pearson's correlation coefficient (r) really show validity and reliability?

Table formatting does not meet the requirements of the IJERPH.

The existing manuscript is not prepared according to the requirements of the IJERPH in terms of its entire structure. There are many discrepancies, especially when authors describe the process of conducting research.

It remains unclear what the authors of the new discovered. What is the relevance and novelty of the study. What are the concrete practical benefits for public health of the results obtained and generalised.

Best Regards

Reviewer 2 Report

I appreciate the great efforts that the authors have made in response to my questions and concerns. However, there are some issues that should be corrected:

  • figure 3 is of poor resolution – it is difficult to read it
  • THIS IS IMPORTANT - Line 279 – ‘moderate validity to a 24-hour recall sur- 279 vey for fruit (r=0.68) and vegetables (r=0.37) ..’ correlation is not a tool for validation. Please correct it. According to the recommendations of Cade et al. (2002), the specific methods should be applied in the validation studies. The analysis of correlation is not the recommended method (so Authors should not conclude on the basis of it). At the same time, the kappa statistic and Bland-Altman method are the recommended methods.
